# Unveiling Influence in Networks: A Novel Centrality Metric and Comparative Analysis through Graph-Based Models

**DOI:** 10.3390/e26060486

**Published:** 2024-05-31

**Authors:** Nada Bendahman, Dounia Lotfi

**Affiliations:** LRIT, Faculty of Sciences, Mohammed V University in Rabat, Rabat 10000, Morocco; d.lotfi@um5r.ac.ma

**Keywords:** social network, graph, influential actor, centrality measure

## Abstract

Identifying influential actors within social networks is pivotal for optimizing information flow and mitigating the spread of both rumors and viruses. Several methods have emerged to pinpoint these influential entities in networks, represented as graphs. In these graphs, nodes correspond to individuals and edges indicate their connections. This study focuses on centrality measures, prized for their straightforwardness and effectiveness. We divide structural centrality into two categories: local, considering a node’s immediate vicinity, and global, accounting for overarching path structures. Some techniques blend both centralities to highlight nodes influential at both micro and macro levels. Our paper presents a novel centrality measure, accentuating node degree and incorporating the network’s broader features, especially paths of different lengths. Through Spearman and Pearson correlations tested on seven standard datasets, our method proves its merit against traditional centrality measures. Additionally, we employ the susceptible–infected–recovered (SIR) model, portraying virus spread, to further validate our approach. The ultimate influential node is gauged by its capacity to infect the most nodes during the SIR model’s progression. Our results indicate a notable correlative efficacy across various real-world networks relative to other centrality metrics.

## 1. Introduction

In the dynamically evolving landscape of digital interconnectedness, social network analysis (SNA) emerges as a crucial disciplinary field, positioned at the intersection of graph theory and sociology. This unique discipline offers a deep understanding of complex interactions among diverse entities such as individuals, organizations, or even URLs [1]. Utilizing nodes to represent entities and links to symbolize their interactions, SNA provides both a visual and mathematical perspective on human interconnections. This analytical approach reveals insights into community dynamics, market trends, and political movements, and finds practical applications in varied areas, ranging from influence marketing to crisis management and public health surveillance. The primary objectives of this analysis include the detection of communities [2,3], the prediction of potential links [4,5,6], and crucially the identification of influential actors [7], thus demonstrating its versatility and relevance in contemporary society.

The significance of identifying these influencers or opinion leaders cannot be understated. These are unique individuals who, even though a minority, can cast vast influence over a majority [8]. Their role becomes even more critical when considering the implications of their influence, such as mitigating rumor spread, virus control [9,10], optimizing energy dissemination [10], and fortifying crucial zones against deliberate threats [7,11,12]. Over time, a rich tapestry of methodologies has evolved to identify influential figures within social networks, each leveraging distinct features such as content and network structure [13]. Content-based detection methods focus on the impact of textual content, considering both linguistic criteria, like the nature of arguments or agreement/disagreement between users, and numerical criteria, such as response frequency, message size, or the extent of relationships. For example, empirical studies have compared messages from influencers with those from non-influencers to discern patterns [14]. Another content-based approach involves analyzing how influencers affect the themes and directions of conversations [15]. Simultaneously, approaches focusing on network structure harness various structural components, with centrality-based methods being particularly prominent [16,17,18]. In these approaches, social networks are typically represented as simple undirected graphs, G = (V, E), where V symbolizes the set of vertices (network users), and E denotes the interconnections between users. We employ centrality measures to capture the structural properties of these nodes. These measures assign real values to nodes, ranking them based on their significance within the network. Structural centrality encompasses two core types: local centrality measures, which are based on the immediate neighborhood of a node [19], and global centrality measures, taking into account a node’s broader membership paths within the network. Local centrality measures include degree centrality [19], local rank [6], and K-shell [20], while global measures feature metrics like eccentricity [21], closeness centrality [19], betweenness centrality [19], and Katz centrality [22]. However, it is crucial to note that despite their utility in understanding network structure, centrality-based methods may exhibit instability in large-scale networks. This is particularly evident in methods like degree centrality, which, while valuable, may overlook nodes with fewer direct connections yet substantial influence within the network’s broader context. This highlights the need for a more nuanced approach to effectively balance both content-based and structural insights to accurately map out the influence landscape within social networks.

While diffusion centrality has garnered attention for its efficacy in modeling the spread of influence, especially in epidemic scenarios, by considering both network topology and information dynamics, this paper introduces a novel methodological approach, centrality degree paths (CDPs), which extends beyond the scope of diffusion centrality. CDPs offer a unique perspective by intricately evaluating the influence exerted by nodes, not only based on their immediate reachability but also by considering the structural paths of influence, encompassing both direct and indirect connections. This nuanced focus allows CDPs to provide a more comprehensive understanding of a node’s influence within the network. Unlike diffusion centrality, which primarily focuses on the potential spread of influence, CDPs delve deeper into the underlying mechanisms of influence propagation, capturing the variability in the lengths of indirect paths and highlighting their significance in shaping network dynamics. By incorporating these structural insights, CDPs offer a more nuanced and detailed perspective on how influence percolates through the network, thereby enhancing our understanding of the dynamics of influence propagation within social networks.

As the field progresses, deep reinforcement learning (DRL) [23] has introduced a novel paradigm in influence maximization, employing iterative learning strategies to optimize the identification of key influencers. Despite the promise of DRL in revolutionizing influence strategies, it encounters notable hurdles, including computational complexity and the extensive data requirements for model training. These challenges, along with additional drawbacks such as training instability, highlight the need for methodologies that are not just efficient but also adaptable across diverse network environments. In this context, the proposed measure aims to address these specific issues by offering a less computationally intensive and more stable alternative for identifying key influencers, thereby ensuring its applicability and effectiveness in various network scenarios.

Addressing these challenges, this paper introduces a novel methodological approach in the field of social network analysis (SNA), standing out for its integration of innovative and unconventional elements. This approach emphasizes the importance of node degree while also considering the broader network context, and is grounded in a solid theoretical foundation. Based on the premise that direct connections (node degree) are significant, our study acknowledges that the true essence of a node’s influence often lies in its broader relationships within the network, including indirect paths of varying lengths. Therefore, rather than solely focusing on the degree of connectivity, we have incorporated the concept of paths of different lengths, aiming for a more nuanced understanding of influence within the network. This innovative approach highlights the importance of indirect paths and their variability in length for a more comprehensive analysis of influence in social networks.

The experimental framework of our study is carefully structured to test the effectiveness of our approach. We have selected a variety of real-world social networks for our analysis, each chosen for its unique characteristics and relevance to our research objectives. Our methodology integrates specific tools and techniques, including advanced statistical methods and computational algorithms, to evaluate the proposed centrality measure. To attest to the viability of our proposed methodology, we subjected it to rigorous testing using Spearman and Pearson correlations across selected real-world social networks. The initial results are promising, with our metric effectively highlighting influential nodes. Continuing our exploration, a cornerstone of our experimental analysis is the application of the susceptible–infected–recovered (SIR) model traditionally used in epidemiology and adapted in our study to simulate the spread of information and influence within social networks. The adaptability of the SIR model to our context demonstrates its utility beyond traditional public health applications, providing valuable insights into the dynamics of information flow and influence propagation in social networks.

The structure of this paper is as follows. Section 2 touches upon prior research, Section 3 details the proposed method, Section 4 offers a deep dive into our findings, and Section 5 wraps up with concluding remarks.

## 2. Related Work

Over the years, social network analysis has continuously evolved, consistently emphasizing the understanding of roles and significance of entities within a network. Among the various aspects studied, centrality computation has stood out as an essential component, having been at the forefront for several decades [24]. This importance stems from the underlying quest to discern and quantify the prominence or influence of individual actors within a broader collective or network. The centrality concept not only gauges the immediate impact of an actor but also reflects its broader implications on the network’s dynamics and flow. As we navigate this section, we will elucidate five fundamental definitions that have been instrumental in shaping the understanding of centrality: degree centrality, closeness centrality, betweenness centrality, PageRank [25] and CNA-GE centrality [26]. Each of these metrics offers a unique perspective on the role and influence of nodes within a network, contributing to the comprehensive landscape of social network analysis.

### 2.1. Degree Centrality

Degree centrality (DC) is foundational in centrality metrics. It quantitatively assesses a node’s importance based on its direct connections within the network. This metric’s intuitive nature correlates increased connections with enhanced influence. Mathematically, using the adjacency matrix $A=(a_{\left\{i,j\right\}})\;$ of an undirected graph G with ***N*** representing the total number of nodes, the degree centrality for a node $v_{i}\in V$ is defined as:(1)$$C^{deg}\left(v_{i}\right)=\frac{1}{N-1}\overset{N}{\underset{j=1}{\sum }}a_{ij}$$
However, while DC offers valuable insights [27], it may have limitations, mainly when employed in complex scenarios such as web page graph analyses. Building on this, a study by Kitsak et al. [28] highlighted an intriguing observation: the most influential nodes are not necessarily the ones with the most connections. Their research led them to explore *k*-core decomposition, an iterative process that segregates nodes based on their minimum degrees [29,30]. A node’s coreness, indicating its rank in the decomposition hierarchy, is directly tied to its capacity to influence network dynamics [31]. In contrast to coreness, the H-index is a local centrality measure that utilizes only partial information, specifically the degrees of the neighbors of the nodes [32]. This emphasizes that while degree centrality offers a foundational understanding, capturing the nuances of influence in complex networks often necessitates a more multifaceted approach.

### 2.2. Closeness Centrality

Closeness centrality is a fundamental metric within the suite of global centrality measures. It is predicated on the assumption that a node’s strategic position within a network is gauged by its global accessibility to all other nodes. For an actor within a social network, high closeness centrality suggests the ability to efficiently interact with others with minimal intermediation, thereby decreasing potential control by others. Mathematically, for an undirected graph G composed of ***N*** vertices, represented by its adjacency matrix A, the closeness centrality ***C^clos^*** of a node ***v_i_*** ∈ ***V*** is given by:(2)$$C^{clos}\left(v_{i}\right)=\frac{N-1}{\sum_{j=1}^{N}dist(v_{i},v_{j})}$$
where ***dist****(**v_i_***,***v_j_***) denotes the shortest distance between the vertices ***v_i_*** and ***v_j_***. The geodesic distance, which is the length of the shortest path connecting the nodes, is commonly employed as the measure of distance in this context, as suggested by Freeman [19]. It is essential to recognize that closeness centrality is only pertinent for strongly connected graphs, where paths exist between all node pairs, ensuring that the network’s global structure contributes to the centrality assessment.

This metric, while informative, assumes a network’s even and uninterrupted connectivity, which may not always be representative of real-world scenarios, where networks can be large, sparse, or fragmented. For instance, in a web graph analysis, where not all pages are equidistant, the utility of closeness centrality could be constrained. Similarly to degree centrality, while providing foundational insights, closeness centrality may not always capture the subtle intricacies of node influence in complex network structures.

### 2.3. Betweenness Centrality

Betweenness centrality (BC) quantifies how often an agent acts as a conduit on the most direct path between two other nodes. Crucial for discerning power dynamics in communication networks, BC measures an entity’s control over information flow. Conceptually, BC offers a probabilistic perspective: it quantifies the likelihood that information traveling between two distinct nodes will traverse through a given node [19]. Formally, the betweenness centrality for any node $v_{i}\in V$ can be articulated as:(3)$$C\left(v_{i}\right)=\overset{N}{\underset{j=1}{\sum }}\overset{N}{\underset{k=1}{\sum }}\frac{g_{jk}\left(v_{i}\right)}{g_{jk}}$$
where $g_{jk}\left(v_{i}\right)$ denotes the number of shortest paths linking nodes $v_{j}$ and $v_{k}$ that incorporate node $v_{i}$, while $g_{jk}$ is the total number of geodesic paths between nodes $v_{j}$ and $v_{k}$. While traditionally attributed to Freeman’s seminal work, the concept of node betweenness centrality also draws on earlier ideas, such as those presented by Anthonisse in 1971 [33], who introduced the notion of “rush in a graph”, which conceptually aligns with what would later be detailed as betweenness centrality. Over time, the concept of betweenness centrality has evolved, incorporating nuances like link betweenness centrality or edge betweenness [34,35]. Before the conceptualization of betweenness centrality, Katz [22] laid the groundwork for understanding network dynamics with an innovative methodology. This approach, distinct from betweenness centrality, prioritizes all potential paths within a network, assigning diminishing significance to increasingly longer paths. By doing so, it effectively captures both the direct and indirect influences within the system, offering a foundational perspective on the importance of network paths that would later complement the development of betweenness and other centrality measures.

### 2.4. PageRank

PageRank, pioneered by Larry Page and Sergey Brin [25], revolutionized web search optimization. It ranks each node based on its connections and affiliations with significant nodes. The foundation of PageRank lies in the “Random Surfer” model, reflecting typical internet navigation patterns. The PageRank formula is given by:(4)$$PR\left(v_{i}\right)=\left(1-\alpha \right)+\alpha \underset{v_{j}\epsilon In\left(v_{i}\right)}{\sum }\frac{PR\left(v_{j}\right)}{Out\left(v_{j}\right)}$$
where $ln\left(v_{i}\right)$ is the set of nodes for which there is a link to $v_{i}$ (i.e., $v_{j}\;\in V$, $\left(v_{i},v_{j}\right)\in E$), $out\left(v_{j}\right)$ is the outgoing degree of $u_{j}$, and α is a damping factor. The computational complexity of the PageRank algorithm mainly arises from the matrix multiplication step, which has a time complexity of *O*(*nm*), where m represents the number of iterations and n represents the number of nodes in the network.

### 2.5. CNA-GE Centrality

CNA-GE centrality, blending traditional network topology with gene expression data, advances node influence analysis in bioinformatics and systems biology. It evaluates node significance not only through structural connections but also by considering the biological data of gene expression levels. The centrality measure [26] for a node, ***v_i_***, can be expressed as:(5)$$C_{CNA-GE}=\deg\left(v_{i}\right)\times (\alpha .expr\left(v_{i}\right)+\beta .\underset{j\epsilon N\left(i\right)}{\sum }expr\left(v_{j}\right))$$
where ***deg***(***v_i_***) is the degree centrality, ***expr****(**v_i_**)* represents the gene expression level at node ***v_i_***, and α and ***β*** are weights modifying the impact of their own and neighbors’ gene expressions, respectively. 

While CNA-GE introduces promising advancements, it also presents unique challenges, particularly in the accurate modeling and interpretation of the biological relevance of centrality scores. The accuracy of CNA-GE centrality heavily relies on the quality and granularity of the gene expression data. Furthermore, applying CNA-GE in non-biological contexts necessitates thoughtful adaptation to ensure that the gene expression component is suitably redefined to align with other types of nodal attributes.

However, another study has led to the development of the “bridging centrality” metric [27], gaining increasing recognition for its proficiency in the analysis of complex networks. It is crucial to emphasize that the effort to combine basic degree metrics with advanced structural determinants introduces its own complexities, making the quest for a universally effective method a formidable challenge. Utilizing various structural properties to identify the most influential nodes in a network indeed proves to be an effective approach. Nevertheless, the selection of these properties for combination remains a challenging task.

Table 1 delineates a comprehensive comparison of CDPs with traditional centrality measures, illustrating how CDPs effectively address the limitations of conventional approaches by integrating advanced structural properties. Each row in the table breaks down a different centrality metric, providing insights into their inherent limitations and the unique benefits introduced by CDPs in assessing both overt and subtle network influences. This holistic view underscores the complexity and effectiveness of combining various network properties to enhance the identification and analysis of influential nodes across diverse network types.

## 3. The Proposed CDP Measure

In the vast field of network analysis, a variety of metrics have been introduced over the years. These measures, while insightful, often operate in isolation, focusing mainly on specific facets of node importance. This section introduces the centrality degree paths (CDPs) measure, a holistic approach that integrates multiple aspects of node influence, in recognition of this limitation. In the subsequent sections, we delve into the intricacies of this innovative approach.

### 3.1. Problem Context

In this research, we delve into the structural analysis of an undirected and unweighted graph, representing a complex social network. Our primary objective is to pinpoint nodes that wield significant influence within this network. This task transcends the simple evaluation of nodes’ degrees, extending to a meticulous examination of paths of varied lengths. Such an approach is critical for encapsulating both direct and indirect interactions between nodes.

The impetus for identifying influential nodes is rooted in the need to comprehend the dynamics of information dissemination, the spread of trends, and the propagation of behaviors across the network. This understanding is pivotal for various applications, ranging from marketing strategies and public health campaigns to the analysis of social movements and the spread of misinformation. This necessitates a nuanced approach that goes beyond traditional analyses, addressing both the observable and the subtle, often overlooked, pathways of influence transmission.

To tackle this challenge effectively, we propose a multifaceted approach. This involves assessing not only the immediate reach of each node, represented by its degree, but also its extended influence, reflected in the network pathways it influences or controls.

The complexity of social networks, with their inherent unpredictability and nonlinear interactions, necessitates a methodology, both simple and robust, capable of capturing the nuances of these networks. Our approach, therefore, focuses on developing a more sophisticated model that considers various factors contributing to a node’s influence. This includes examining network topology, node centrality, and the role of clusters within the network.

While existing measures like eigenvector centrality and length-scaled betweenness also consider the impact of path lengths, CDPs take into consideration both direct connections (node degrees) and measure the impact of indirect paths through specific calculations that account for the variability in path length and their influence on determining a node’s centrality. This nuanced approach emphasizes the impact of both direct and extended network connections, thereby allowing CDPs to capture influence dynamics more comprehensively and address both overt and subtle influence pathways within the network.

Furthermore, recent works have explored the application of machine learning techniques to enhance network centrality measures [26,36], indicating a growing interest in advancing the field through innovative methodologies. Li et al. [37] have also delved into centrality learning, introducing novel approaches to capture the nuances of influence within networks. Each one of the previously proposed measures can be said to leverage network structures and paths to assess influence uniquely.

### 3.2. CDP Measure

Traditional local and global centrality metrics, while insightful, have inherent limitations, particularly when applied to specific network typologies. The amalgamation of multiple measures can provide enriched insights. In this research, we introduce the CDP score, a composite measure that accentuates degree centrality while incorporating the number of paths. Solely relying on degrees may result in potentially overlooking crucial network dynamics. The proposed approach enhances node importance using squared degrees and concurrently considers the number of simple paths. Let $G=\left(V,E\right)$ represent an undirected simple graph.

The CDP centrality score of a node x∈V is defined as follows:(6)$$\mathrm{CDP}x=\frac{\deg\left(x\right)2}{\left|P{\left(x,y\right)}^{l\leq d}\right|}$$
where $\left|P{\left(x,y\right)}^{l\leq d}\right|$ represents the aggregate count of all paths from node *x* to any other node *y* within the network, where the path length *l* does not exceed the predefined maximum *d*. This variable *d* acts as a control to limit the path length, ensuring that both direct and extended influences are captured. The variable *y* iterates over all nodes reachable from *x* under these conditions, allowing a comprehensive evaluation of *x*’s influence.

To derive the CDP centrality score for nodes in a graph, the methodology below is proposed.

Squared Degree of a Node (*deg*(*x*))^2^: The degree of a node in a network graph denotes the count of edges connected to that node. Squaring this value accentuates the impact of nodes with heightened degrees over those with lesser degrees, providing a balance that sufficiently recognizes the influence of highly connected nodes without disproportionately amplifying their impact. This moderation prevents the overshadowing of other structural features of the network. By squaring the degree, the formula magnifies the significance of highly interconnected nodes more than a linear scaling would, yet avoids the extreme influence escalation that would result from higher powers.Path Count ($\left|P{\left(x,y\right)}^{l\leq d}\right|$): This element of the formula calculates the number of paths between nodes *x* and *y*, with the condition that only those paths of length *l* that are less than or equal to a predefined distance *d* are included. This distinction is crucial, as it considers both the direct influence of a node (reflected by its degree) and its indirect influence via network connections within a determined range. This approach provides a more comprehensive view of a node’s capacity to disseminate influence across the network.Division of the Two Components: The division of the squared degree of a node by the count of paths within a stipulated distance appears to standardize the influence exerted by a node. This serves to mitigate the potential distortion caused by nodes with exceptionally high degrees, particularly in extensive or dense networks.

Collectively, the choice to square the degree underscores nodes that are not merely connected but highly interconnected, conceivably serving as hubs or pivotal influencers in the network. The incorporation of path count introduces a layer of intricacy to the analysis, acknowledging that influence within a network extends beyond direct connections to encompass the dissemination of these connections throughout the network. Such methodological deliberations enhance the scholarly rigor of the manuscript by furnishing a nuanced and comprehensive measure of node influence in network analysis.

This procedural approach facilitates the identification of pivotal nodes, which can be visualized and verified as illustrated in Figure 1.

### 3.3. Example

To illustrate the application of the CDP score, we consider Figure 1. For this analysis, a specific choice of *l* = 2 has been made, focusing on direct neighbors and those of second-degree proximity. This parameter choice provides a balance between depth and computational feasibility.

The CDP score in Table 2 displays the influence of each node. As depicted, node 3 stands out as the most influential despite its degree, underscoring the advantage of the CDP measure.

To underscore the added value of the proposed metric, we turn our attention to the network depicted in Figure 1. When relying solely on degree centrality, node 5 stands out as the most influential with a degree of 4, followed by nodes 1, 3, 8, and 11, each having a degree of 3. However, when applying the CDP metric, node 3 emerges as the predominant node with a score close to 1.8. It is followed by node 5 with a score of 1.77 and node 8 with 1.5. Even though node 3 has a lower degree than node 5, its position within the network makes it more influential. In summation, this innovative metric elevates accuracy in ranking by assimilating not just the degree centrality but also the abundance of simple paths throughout the network.

## 4. Experiments

In the previous section, we introduced and discussed the conceptual framework of our novel CDP method. Building on that foundation, this section is devoted to an empirical evaluation of the proposed method. We employ seven distinct real-world networks for this purpose: Zachary’s Karate Club network, Dolphin network, Les Misérables network, US Politics Books network, American College Football, USAir97, and Mouse-Kasthuri network. Our primary objective is to ascertain the correlation between traditional centrality measures and the results derived from the proposed CDP method. To rigorously evaluate this association, both Pearson’s and Spearman’s correlation metrics have been employed, offering a comprehensive understanding of the statistical interdependence between the centrality metrics and the scores generated by the CDP approach. Furthermore, the propagation dynamics within these networks are examined using the susceptible–infected–recovered (SIR) model, enabling us to gauge the spreading efficiency associated with each node. The performance of the CDP method, in conjunction with the SIR model, is further scrutinized using Kendall’s tau and overlap coefficients.

### 4.1. Dataset

To critically assess our proposed metric, we engage with an array of networks originating from varied disciplines. Our scrutiny is meticulously confined to undirected simple graphs, thus ensuring an unambiguous omission of loops and duplicate edges. An outline of the datasets marshaled for this exploration is presented subsequently. In Table 3, we utilized the dataset previously introduced in [38] and incorporated two additional large datasets (USAir97 and Mouse-Kasthuri) to further validate the performance of the proposed CDP metric.

Zachary’s Karate Club: A representation of social interactions among 34 members of a karate club at a US-based university during the 1970s.Dolphin: A depiction of regular interactions among 62 dolphins in a community near Doubtful Sound, New Zealand.Les Misérables: A network delineating the coappearances of characters in Victor Hugo’s celebrated novel *Les Misérables*.US Politics Books: This network visualizes the co-purchasing patterns of political books around the time of the 2004 US presidential election, as recorded by Amazon.com.American College Football: A matrix of football games amongst Division IA colleges during the Fall 2000 season.USAir97: A topological representation of the US air travel landscape in 1997, with nodes as airports and edges as direct flight connections.Mouse-Kasthuri: A dataset from the Neurodata repository, representing unweighted fiber tracts connecting vertices in brain networks.

### 4.2. Evaluation Metrics

#### 4.2.1. Pearson’s Correlation

Denoted by ρ, Pearson’s correlation quantifies the linear association between two variables. This coefficient, often used in linear regression analysis, ranges from −1 to +1. A coefficient of +1 implies a perfect positive linear relationship, −1 denotes a perfect negative linear relationship, and a value of 0 suggests no linear correlation. The formula for calculating Pearson’s correlation for variables *x* and *y* is:(7)$$\rho_{x,y}=\frac{cov\left(x,y\right)}{\sigma_{x}\sigma_{y}}$$
where *cov*(*x,y*) represents the covariance between *x* and *y*, while $\sigma_{x}$ and $\sigma_{y}$ denote the standard deviations of *x* and *y*, respectively.

#### 4.2.2. Spearman’s Correlation

This nonparametric measure assesses the strength and direction of the monotonic relationship between two variables. Distinct from Pearson’s correlation, which evaluates linear relationships, Spearman’s correlation captures both linear and nonlinear monotonic associations. It computes the correlation based on the rank values of the variables, rather than their actual values. A coefficient value of +1 or −1 reflects a perfect monotonic relationship. The formula for Spearman’s correlation is given by:(8)$$\rho_{R\left(x\right),R\left(y\right)}=\frac{cov\left(R\left(x\right),R\left(y\right)\right)}{\sigma_{R\left(x\right)}\sigma_{R\left(y\right)}}$$
In this equation, *ρ* signifies the Pearson correlation coefficient when applied to rank variables, *cov*(*R(x*),*R*(*y*)) is the covariance of the rank variables, and $\sigma_{R\left(x\right)}$ and $\sigma_{R\left(y\right)}$ represent their respective standard deviations.

#### 4.2.3. SIR Model

The susceptible–infected–recovered (SIR) model is a well-known method for evaluating centrality measures in network analysis. This epidemiological model simulates the spread of a virus within a network, categorizing nodes into three distinct states: susceptible (*S*), infected (*I*), and recovered (*R*) [43].

Susceptible (*S*): A susceptible node is one that is unaware of the information spreading within the network. These nodes are healthy but not immune, and can be infected by adjacent infected nodes. Initially, all nodes are considered susceptible except for the source node.Infected (*I*): An infected node has received and is aware of the information propagating through the network. It actively transmits this information to its neighbors. An infected node transitions to the recovered state after a certain period, dictated by an infection probability *β* at each time step and a recovery probability λ*dt* within any time interval *dt*. The average duration of a node being infected is represented by D.Recovered (*R*): Recovered nodes have lost interest in the information and no longer spread it. They also become immune to further infection. At the end of the process, only susceptible and recovered nodes remain in the network.

The dynamics of the SIR model are governed by a set of ordinary differential equations, describing the transitions between these states. The total population in the network is denoted by *N*, and at any time *t*, the sum of susceptible, infected, and recovered nodes equals *N*. The SIR system can be described by the following system of ordinary differential equations:
$$\frac{dS}{dt}=\frac{\beta IS}{N}$$
$$\frac{dI}{dt}=\frac{\beta IS}{N-\lambda I}$$
$$\frac{dR}{dt}=\lambda I$$
(9)S(t)+I(t)+R(t)=N
The SIR model’s utility extends beyond traditional epidemiological contexts to the analysis of information spread in social networks, providing insights into the influence exerted by various nodes based on their centrality measures.

#### 4.2.4. Kendall’s Tau

Denoted by *τ*, Kendall’s tau is a statistic that measures the rank correlation between two variables, assessing the association based on their ranks. It is particularly useful for nonlinear relationships. The pairs of observations ($x_{i}$,$\;y_{i}$) and ($x_{j}$,$\;y_{j}$) are concordant if both $x_{i}$ and $x_{j}$ or $y_{i}$ and $y_{j}$ are either both increasing or both decreasing. They are discordant if one is increasing and the other is decreasing. Kendall’s tau is calculated as the difference between the probability of concordant pairs and discordant pairs. The value of *τ* ranges from −1 (perfect disagreement) to +1 (perfect agreement), with 0 indicating no correlation. Kendall’s tau is given by:(10)$$\tau =\frac{\left(\mathrm{number}\;\mathrm{of}\;\mathrm{concordant}\;\mathrm{pairs}-\mathrm{number}\;\mathrm{of}\;\mathrm{discordant}\;\mathrm{pairs}\right)}{\frac{1}{2}n\left(n-1\right)}\;$$
where *n* is the number of observations.

#### 4.2.5. Overlap Coefficient

This similarity measure, related to the Jaccard index, quantifies the overlap between two finite sets. Represented as the size of the intersection divided by the smaller sizes of the two sets, the overlap coefficient is a valuable tool for comparing different sets. It is particularly useful for assessing the correlation of the most influential node sets obtained from ranking scores and the SIR model. The overlap coefficient ranges from 0 (no overlap) to 1 (complete overlap), with higher values indicating more significant overlap. The formula for calculating the overlap coefficient between two sets *X* and *Y* is given by:(11)$$overlap\left(X,Y\right)=\frac{\left|X\cap Y\right|}{\min\left(\left|X\right|,\left|Y\right|\right)}\;$$
In this context, a higher overlap coefficient value indicates greater reliability of the ranking score in identifying influential nodes within the network.

### 4.3. Experimental Results

#### 4.3.1. Top-Ranked Nodes

In Table 4, we delineate the results of CDPs in comparison with degree centrality, closeness centrality, betweenness centrality, PageRank centrality, and CNA-GE centrality for six pivotal nodes across seven real-world networks. We leverage the findings and results from our previous work [38]. Additionally, we present new results from two additional datasets, USAir97 and Mouse-Kasthuri, providing further insights into the proposed CDP score.

In networks like Dolphin and Zachary’s Karate Club, the highly influential nodes identified by the CDP score closely resemble those obtained through methods such as degree centrality and PageRank. Moreover, in networks such as Zachary’s Karate Club and Dolphin, CDPs excel not only in recognizing nodes with the highest direct connectivity (degree centrality) or those acting as critical bridges (betweenness centrality), but also nodes that influence through a combination of direct links and multiple indirect pathways. This approach is particularly beneficial in complex networks where influence depends not only on direct connections or the shortest paths. For example, in the Les Misérables network, nodes central to the narrative structure may not have the highest degree centrality, but they are crucial for linking different groups of characters through various story arcs. CDPs better capture this nuanced role by integrating the number of pathways of different lengths, which traditional measures may overlook.

Turning our gaze to more complex networks like USAir97 and Mouse-Kasthuri, the CDP method’s adaptability is underscored. In the USAir97 network, the nodes highlighted by CDPs exhibit a significant overlap with those identified by PageRank, showcasing the method’s consistent performance across varying network topologies. The Mouse-Kasthuri dataset further accentuates CDP precision, as it harmoniously aligns with both the PageRank and degree centrality measures, indicating its capability to adeptly navigate intricate neural networks. Collectively, these aspects highlight the CDP method’s ability to identify influential nodes in contexts where node importance goes beyond simple direct connectivity, thus offering a more comprehensive perspective of network dynamics.

#### 4.3.2. Pearson and Spearman Results

In our quest to validate the efficacy of the newly proposed CDP score, comprehensive evaluations were undertaken, juxtaposing it with prevailing centrality metrics—degree centrality (DC), closeness centrality (CC), betweenness centrality (BC), PageRank centrality and CNA-GE centrality—across a diverse spectrum of benchmark networks. The ensuing correlations, both Pearson and Spearman, have been meticulously delineated in Table 5 and Table 6. Additionally, to further substantiate our findings and enhance the robustness of our analysis, we incorporated results from our previous work [38]. Moreover, we introduced new results derived from two additional large datasets (USAir97 and Mouse-Kasthuri), aiming to confirm the effectiveness of the CDP score in diverse network scenarios. This comprehensive approach provides a more holistic understanding of the proposed metric’s performance and applicability.

When we delve into the intricacies of these correlations, certain salient observations emerge. First and foremost, a compelling correlation between the CDP score and the DC method can be discerned. Such an association intimates that the CDP score, in its essence, is as adept at echoing similar facets of node centrality as those evoked by degree centrality. Furthermore, our results exhibit a heightened reliability when correlated with the PageRank method. This can be attributed to their shared foundational principles, specifically the emphasis on node degree and the intricate interplay of links between origin and terminal nodes, operationalized through the count of paths spanning a length of two. However, when juxtaposed with the BC method, the correlation is relatively attenuated. This stems from the intrinsic nature of the BC method, which inherently skews towards nodes that find themselves integral to a more extensive gamut of paths within the network. Contrarily, the CDP method has been conceived so as to mediate harmoniously between node degree and betweenness centrality, ensuring neither is disproportionately favored. Although the correlation between CDPs and closeness centrality is more moderate, this reflects the distinctive capability of the CDP measure to capture extended dimensions of influence beyond immediate proximity, thereby providing a broader perspective of centrality in complex networks.

The NCA-GE centrality correlations provide further insight, notably exhibiting a higher correlation in the Dolphin network, suggestive of a robust rank correlation in contexts where biological data play a significant role. Conversely, moderate correlations observed in less biologically inclined networks suggest the potential need for adapting NCA-GE centrality to better accommodate networks where gene expression is not a primary factor.

#### 4.3.3. The Influence of Parameter *l*

The proposed CDP measure’s performance, when adjusted against varied path length “*l*” values, the performance of the proposed CDP measure presents compelling insights. Using an expansive dataset set that comprises Dolphins, Les Misérables, US Politics Books, American College Football, USAir97, and Mouse-Kasthuri, we juxtaposed the correlation against mainstream metrics, with “*l*” spanning from 2 to 4 using the Pearson coefficient.

We incorporated insights from our previous work to further enrich our analysis [38]. We augmented our analysis by including the performance of the CDP score with the variation in path length for the networks USAir97 and Mouse-Kasthuri. This addition provides a more comprehensive understanding of how the CDP measure behaves in diverse network structures, further enriching our insights into its adaptability and effectiveness.

Upon assessing the PageRank metric, a clear pattern emerges: the “*l*” value predominantly shines at 2 for most datasets, echoing our foundational findings. For instance, networks like Dolphins, US Politics Books, American College Football, and Mouse-Kasthuri showcase correlation values of 0.976, 0.97, 0.966, and 0.983, respectively, when “*l* = 2.” Conversely, the Les Misérables dataset and USAir97 stand out, with the former reaching its zenith at 0.96 for “*l* = 3” and the latter peaking at 0.893 for “*l* = 2.” A parallel trend is discernible in the degree centrality metric. With Dolphins, US Politics Books, and Mouse-Kasthuri, the optimal “*l*” gravitates towards 2, exhibiting values of 0.966, 0.94, and 0.964, respectively. However, Les Misérables veers off this course, registering an optimal correlation of 0.864 at “*l* = 3.” USAir97’s best result of 0.901 is also at “*l* = 2”, fitting the broader trajectory. The betweenness metric further enriches this narrative. With Dolphins, a peak correlation of 0.914 is noted at “*l* = 2”, in sync with prior revelations. However, the best correlations for datasets like American College Football, Les Misérables, USAir97, and Mouse-Kasthuri are 0.327, 0.914, 0.818, and 0.666, respectively.

Interestingly, while “*l* = 2” maintains its dominance, Les Misérables diverges from the trend, solidifying its optimal “*l*” value at 3. On the other hand, the Zachary’s Karate Club network stands out with its unique optimal “*l*” values, which are exclusively 3 and 4. A salient aspect underpinning this analysis is the conceptual depth offered by paths of “*l* = 2”, which unveils information about secondary neighbors (as illustrated in Figure 2). Such granularity ensures that the CDP measure adeptly captures the nuances of node influence within the networks.

To summarize, the prevailing consensus advocates for “*l* = 2” as the premier path length for the CDP measure across various networks. Nonetheless, datasets such as Les Misérables and Zachary’s Karate Club necessitate a nuanced approach, hinting at “*l* = 3” or “*l* = 4” for optimal efficacy.

#### 4.3.4. SIR Model Results

In our quest to highlight the efficacy of the newly introduced CDP score, we leverage the susceptible–infected–recovered (SIR) model—a paradigm simulating viral dissemination across networks. Our procedure entails determining the spreading efficiency for every network node. The ultimate goal is a side-by-side comparison of the CDP metric with benchmark centralities to rank node spreading efficiencies, subsequently spotlighting influential propagators. In this experimentation phase, an initial ranking of nodes within each network was orchestrated using established metrics: degree centrality (DC), closeness centrality (CC), betweenness centrality (BC), PageRank, CNA-GE centrality, and the proposed CDP metric. We then employed the SIR model, treating each node as an information fountainhead (akin to an infection source within the SIR framework). This method calculates the resultant infected nodes post-process. We fixed the number of iterations for our study’s parameters at 50. The specific beta value for every real-world network is enumerated in Table 3, with *λ* set at 1.

Subsequently, we assessed Kendall’s tau correlation juxtaposing the node ranking, as determined by centrality measures, against the infected node count from the SIR model, considering all nodes within the network for this analysis, where “infected node count” refers to the total number of nodes classified as “infected” at the end of the simulation. The results are tabulated in Table 7.

Upon inspecting Table 7, it is palpable that the CDP metric’s performance is consistently formidable across many network datasets. The CDP score remains competitive across all datasets, displaying minimal fluctuations. This is a testament to its robustness and adaptability to diverse network structures. In networks like American College Football, Les Misérables, and US Politics Books, the CDP method slightly edges out the PageRank algorithm. While the margins are nuanced, this demonstrates the potency of the CDP method in certain contexts, indicating its potential utility for researchers and practitioners alike. For datasets like USAir97 and Mouse-Kasthuri, which are arguably representative of more intricate real-world systems, CDPs exhibit a close alignment with other well-regarded metrics like betweenness and degree centrality. This parallelism is especially commendable given the complexity inherent in such networks.

One of the hallmarks of an effective metric is its stability across a variety of scenarios. The CDP metric’s performance in networks like Dolphin, Zachary’s Karate Club, and Les Misérables highlights its capability to maintain a steady ranking, which is crucial for accurate node assessments. While degree and betweenness centralities have peaks in specific networks, their performance is inconsistent with the CDP method. PageRank, a widely recognized algorithm, is at times outperformed by CDPs, emphasizing the latter’s potential as a primary tool in network analysis.

Table 7 underscores the contrast between CDPs’ robust performance across diverse networks and the limitations of CNA-GE centrality, particularly in non-biological contexts. CDPs consistently outperform even well-established metrics like PageRank in various scenarios, highlighting the method’s versatility and reliability. Conversely, CNA-GE’s negative correlations in some datasets reveal its challenges when applied beyond its specialized domain, suggesting a need for careful adaptation to align with general network dynamics. This highlights CDPs’ superior adaptability and effectiveness as a centrality measure across different network structures.

In Table 8, we engage the overlap coefficient [44] as an analytical lens to evaluate the congruence between the ranking scores and the top 10 influential nodes delineated by the SIR model. This methodological approach is particularly germane, as it gives us a tangible metric to quantify the intersection of results between different methods. A cursory observation of Table 8 reveals a rather compelling narrative in favor of the CDP metric it distinctly differentiates itself in networks such as American College Football and Zachary’s Karate Club. In the American College Football network, it is noteworthy that while other metrics hover at a 0.7 or lower overlap coefficient, CDPs boast a perfect 1. This indicates that CDPs have flawlessly identified the top 10 influential spreaders in this network as per the SIR model, a feat unparalleled by other metrics. In networks like US Politics Books, Les Misérables, and Dolphin, the CDP metric holds its ground, paralleling the performance of established centrality measures such as PageRank, DC, and BC. This is an attestation of CDPs’ generalizability and robustness across diverse networks. The Zachary’s Karate Club network offers an illustrative case study. While degree, betweenness, and PageRank all score a 0.3 overlap coefficient, CDPs marginally advance with 0.4. This incremental advantage might appear subtle but signifies CDPs’ nuanced capabilities that the other metrics might overlook. CNA-GE, notably in the Dolphin, Les Misérables, and US Politics Books networks, underperforms compared to CDPs, demonstrating only minimal overlap with the top influencer nodes identified by other metrics. The modest overlap coefficients for CNA-GE indicate its potential limitations in accurately capturing key spreader nodes across diverse networks, underscoring the need for tailored applications or methodological adjustments in non-biological contexts. This analysis further reinforces the adaptability and precision of CDPs in effectively pinpointing influential nodes across different types of networks.

Synthesizing the information, it is clear that the CDP method is not merely an addition to the suite of centrality measures available to analysts—it is a significant enhancement. Its remarkable performance across diverse networks and consistent ranking ability underscores its potential as a vital tool for future network analyses. It effectively bridges the gap between established methods and the ever-evolving complexities of real-world networks, showcasing promise as an influential spreader identifier. The CDP method exemplifies the vanguard of centrality measures, bringing with it a refined precision and an adeptness that in certain networks even surpasses time-tested metrics. The data underscore its prowess in ranking nodes with high fidelity and spotlighting influential spreaders with a commendable degree of accuracy.

## 5. Conclusions

In the intricate tapestry of network analysis, the study presented herein sought to unveil a novel approach, the CDP method, to identify influential nodes within varied real-world networks. Our comprehensive assessment and juxtaposition with established centrality measures—degree centrality, closeness centrality, betweenness centrality, PageRank and CNA-GE centrality—illuminate the adeptness and precision of the CDP technique. The CDP method consistently showcased its versatility from simpler networks like Dolphin and Zachary’s Karate Club to the more complex USAir97 and Mouse-Kasthuri. This ability to adapt and maintain precision, especially in complex structures, underscores its potential as a pivotal tool for future network investigations. Moreover, the convergences with and occasional divergences from traditional methods provide a fresh perspective, reminding us of the continual evolution of network analysis paradigms. As we stand at the cusp of expanding digital interconnectedness, the CDP method presents a valuable methodology for achieving deeper, more nuanced insights. In conclusion, while this study has paved the way for a more holistic understanding of network structures, it also beckons further exploration. We remain hopeful that the CDP method will be instrumental in unraveling the myriad mysteries of networks, driving both academic and practical advancements in the domain.

## Figures and Tables

**Figure 1 entropy-26-00486-f001:**
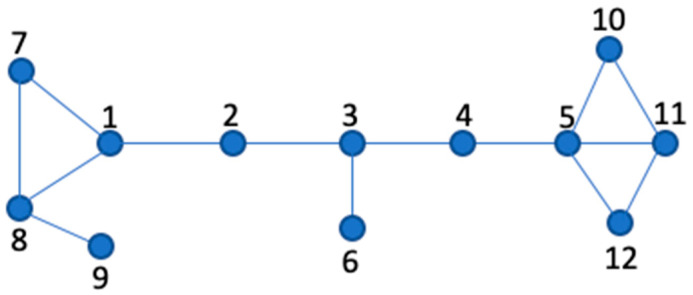
A graph representing a social network (G); nodes correspond to individual users, numbered for identification, and edges represent social relationships between them.

**Figure 2 entropy-26-00486-f002:**
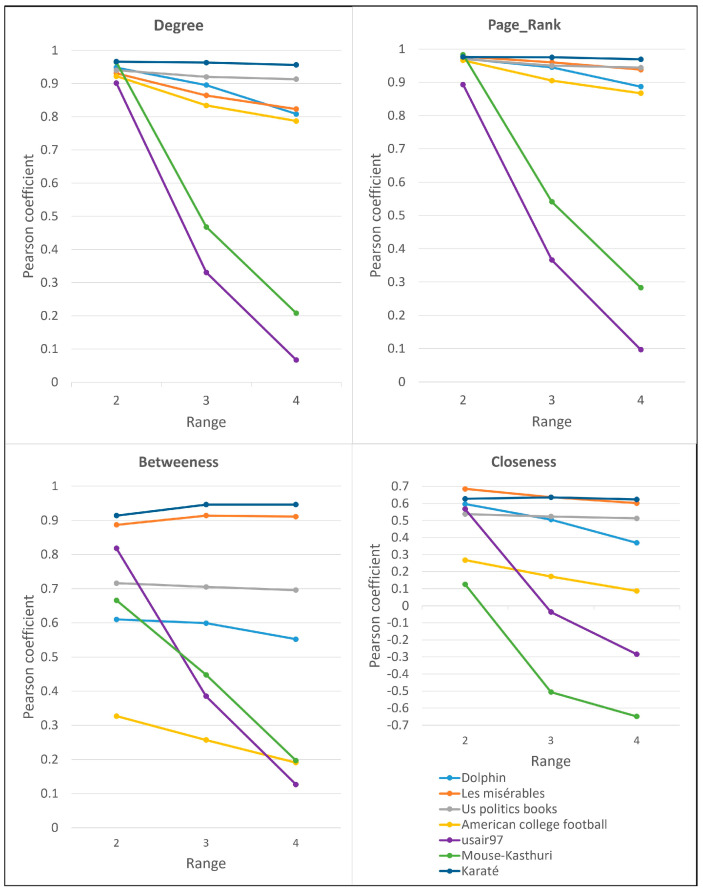
Performance of CDPs using different values of length *l* in the range from 2 to 4.

**Table 1 entropy-26-00486-t001:** Comparative Analysis of Centrality Measures and the Enhanced Approach of Centrality Degree Paths (CDPs).

Concept	Other Centrality Measures	Description and Limitations in Other Measures	Incorporation in CDPs	Advantages of CDPs
Direct and Indirect Connections	Degree Centrality (DC) [19]	Calculates importance based only on the number of direct links, neglecting indirect interactions.	CDPs balance direct connections and indirect paths to offer a holistic view of node influence.	CDPs identify influential nodes not evident by traditional measures, enhancing network understanding.
Network Scale Sensitivity	Closeness Centrality [19]	Assesses how quickly a node can reach all other nodes, often impractical in large networks due to computational complexity.	CDPs adjust this approach by considering optimal paths and evaluating impact on larger network scales.	CDPs allow for more practical application in vast networks without compromising result accuracy.
Granular Path Analysis	Betweenness Centrality (BC) [19]	Primarily focuses on shortest paths, which may not capture all influential pathways within a network, especially in complex or large-scale networks.	CDPs apply a unique weighting to paths, emphasizing both direct and extended network connections, and consider all path lengths, rather than just the shortest.	CDPs provide a more granular and comprehensive analysis of influence within the network, capturing both overt and subtle influence pathways that betweenness centrality might overlook.
Multi-Step Influence Propagation	PageRank [25]	Well suited for citation or web networks, but less effective in social networks where interactions are less hierarchical.	CDPs evaluate the cumulative effects of interactions through multiple steps, allowing for a better estimation of actual influence.	CDPs increase accuracy in social networks and other network types where influence dynamics are complex.
Nonlinear Influence Dynamics	Eigenvector Centrality (EC)	Importance is based on connectivity to other influential nodes, but can be limited by concentrating influence.	CDPs incorporate path length considerations to modulate influence based on actual connectivity, including indirect ones.	CDPs ensure a more nuanced understanding of influence by accounting for nonlinear effects and indirect connections.
Non-Uniform Interactions	Katz Centrality [22]	Gives importance to all paths, but with exponential decay based on length, potentially missing relevant paths.	CDPs customize path weighting based on the variability and contextual importance of interactions.	CDPs tailor influence measurement to more accurately reflect the specific structure and dynamics of the network.

**Table 2 entropy-26-00486-t002:** CDP and degree scores.

*x*	1	2	3	4	5	6	7	8	9	10	11	12
CDPs	1.28	0.66	1.8	0.57	1.77	0.33	0.66	1.5	0.33	0.57	1.12	0.57
Degrees	3	2	3	2	4	1	2	3	1	2	3	2

**Table 3 entropy-26-00486-t003:** Fundamental topological features of benchmark real-world networks. Within this framework, *|V|* represents the total count of nodes, *|E|* indicates the total number of edges, *k* stands for average degrees, *C* is the clustering coefficient, *r* signifies the assortative coefficient, and *β* denotes the infection probability.

Network	*|V|*	*|E|*	*C*	*r*	*k*	*β*
Zachary’s Karate Club [39]	34	78	0.5706	−0.475	4.588	0.12
Dolphin [40]	62	159	0.258	−0.0435	5.129	0.14
Les Misérables [41]	77	254	0.74	−0.165	3.3	0.082
US Politics Books [42]	105	441	0.48	−0.12	8.4	0.083
American College Football [34]	115	613	0.40	−0.162	10.66	0.093
USAir97 [3]	332	2126	0.625217	−0.207876	12.8072	0.022
Mouse-Kasthuri [3]	1000	1700	0	−0.215013	3	0.023

**Table 4 entropy-26-00486-t004:** The top six nodes ranked using CDPs, DC, BC, and PageRank methods.

	Dolphin						Zachary’s Karate Club					
Rank	1	2	3	4	5	6	1	2	3	4	5	6
Degree	15	46	38	52	34	58	34	1	33	3	2	32
Betweenness	37	2	41	38	8	18	1	34	33	3	32	9
Closeness	37	41	38	21	15	2	1	3	34	32	33	14
PageRank	15	18	52	58	38	46	34	1	33	3	2	32
NCA-GE	15	38	46	34	51	41	1	34	3	33	9	14
CDPs	52	15	18	58	46	38	34	1	33	2	3	4
	US Politics Books						American College Football					
Rank	1	2	3	4	5	6	1	2	3	4	5	6
Degree	12	8	84	3	72	73	104	88	67	53	15	7
Betweenness	30	49	9	12	72	76	82	0	80	58	38	69
Closeness	30	58	49	7	9	76	58	80	88	106	6	92
PageRank	12	8	3	84	72	66	5	1	3	0	6	104
NCA-GE	8	12	84	73	30	3	2	67	53	15	7	88
CDPs	12	8	3	72	84	66	5	1	3	0	43	18
	Les Misérables						USAir97					
Rank	1	2	3	4	5	6	1	2	3	4	5	6
Degree	11	48	55	27	25	58	117	260	254	151	181	229
Betweenness	11	0	48	55	23	25	117	7	260	200	46	181
Closeness	11	55	27	25	48	58	117	260	66	254	200	181
PageRank	11	0	48	55	27	25	117	260	200	46	44	254
NCA-GE	11	48	55	25	27	58	117	260	254	181	66	165
CDPs	11	0	48	55	23	27	117	260	254	12	151	181
	Mouse-Kasthuri											
Rank	1	2	3	4	5	6						
Degree	6	83	92	0	35	218						
Betweenness	6	83	92	0	35	218						
Closeness	6	83	92	0	147	561						
PageRank	6	83	35	92	0	218						
NCA-GE	6	147	210	561	111	588						
CDPs	6	83	35	0	92	218						

**Table 5 entropy-26-00486-t005:** Pearson correlations comparing established centrality measures with CDPs.

Score	Zachary’s Karate Club	Dolphin	Les Misérables	US Politics Books	American College Football	USAir97	Mouse-Kasthuri
Degree	0.966	0.948	0.931	0.940	0.922	0.901	0.964
Betweenness	0.914	0.61	0.887	0.716	0.327	0.818	0.90
Closeness	0.628	0.597	0.686	0.538	0.268	0.568	0.126
PageRank	0.976	0.971	0.977	0.970	0.966	0.893	0.983
NCA-GE	0.703	0.829	0.796	0.819	0.819	0.747	0.119

**Table 6 entropy-26-00486-t006:** Spearman correlations comparing established centrality measures with CDPs.

Score	Zachary’s Karate Club	Dolphin	Les Misérables	US Politics Books	American College Football	USAir97	Mouse-Kasthuri
Degree	0.86	0.94	0.93	0.81	0.87	0.811	0.458
Betweenness	0.80	0.81	0.70	0.74	0.33	0.774	0.39
Closeness	0.673	0.621	0.487	0.500	0.245	0.426	−0.321
PageRank	0.91	0.96	0.94	0.87	0.98	0.845	0.69
NCA-GE	0.459	0.832	0.776	0.571	0.566	0.525	−0.449

**Table 7 entropy-26-00486-t007:** Kendall’s correlations of centrality methods and spreading efficiency.

	Degree	Betweenness	Closeness	PageRank	NCA-GE	CDPs
Dolphin	0.783	0.784	0.475	0.778	0.052	0.775
Zachary’s Karate Club	0.746	0.759	0.524	0.729	0.009	0.701
American College Football	0.83	0.79	0.485	0.78	0.102	0.79
Les Misérables	0.74	0.77	0.447	0.73	−0.086	0.74
US Politics Books	0.731	0.744	0.525	0.725	−0.135	0.736
USAir97	0.75	0.77	0.461	0.743	0.007	0.743
Mouse-Kasthuri	0.706	0.699	0.004	0.644	0.0216	0.644

**Table 8 entropy-26-00486-t008:** Overlap coefficient of the top 10 influential spreaders.

	Degree	Betweenness	Closeness	PageRank	NCA-GE	CDP
Dolphin	0.2	0.2	0.2	0.2	0.1	0.2
Zachary’s Karate Club	0.3	0.3	0.3	0.3	0.4	0.4
American College Football	0.7	0.2	0.2	0.7	1	1
Les Misérables	0.2	0.2	0.1	0.2	0.1	0.2
US Politics Books	0.2	0.2	0.2	0.2	0.0	0.2

## Data Availability

The data presented in this study are available on request from the corresponding author.

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
