# Peer review of "Unveiling Influence in Networks: A Novel Centrality Metric and Comparative Analysis through Graph-Based Models"

_entropy, 2024, doi:10.3390/e26060486_

Round 1

Reviewer 1 Report

Comments and Suggestions for Authors

This article proposes a centrality measure called the Centrality Degree Pagerank (CDP). 

Briefly, I am unfamiliar with this specific combination of the degree and PageRank measures. However, the current article fails to convince that (1) the proposed measure is novel enough, provided the variety of published combinations of centrality measures, and (2) the proposed measure is useful.     

The article cites the theoretical taxonomy of centrality measures developed by Everett and Borgatti however, (1) Their work is cited [29] on a side claim referring to betweenness whereas there are more appropriate works specifically discussing the probabilistic view of BC. (2) The categorization of centrality measures used by the authors is not aligned with the theoretical classifications used by Everett and Borgatti. The differences are not discussed and the proposed categorization is not justified in light of [29]. 

Diffusion centrality (Kang et al. 2016 and subsequent publications) seems to be a better fit to maximize the influence of epidemic seeds. Why is CDP better?

Betweenness centrality is frequently attributed to Anthonisse 1971 rather than Freeman 1978. Are the authors sufficiently familiar with the respective history to claim whose brainchild it is? If not, the paper should contain refrain from ungrounded claims. If yes, it should provide evidence more significant than similar allegations in other articles. The same is true for attributing BC to Anthonisse. This reviewer is not qualified to judge which attribution is more correct. 

Katz [19] cannot broaden the horizon innovating over betweenness because it was published twenty years before betweenness.

This is not the first work that provides an amalgamation of centrality measures. There are multiple works providing optimal combinations of centrality measures using traditional machine learning or employing graph neural networks to learn new centrality measures from scratch.

Networks used in the experiments are very small. Considering the computational complexity of the proposed measure and the fact that BC and PR can be computed today on very large networks, there is no reason to ignore complex networks with 10K-100K nodes in the experiments.

Correlation is not sufficient to prove the effectiveness of a centrality measure. Since all centrality measures are correlated, it will only show that CDP is a kind of centrality measure but not that it is better than others for a particular task.

There are no citations of the overlap coefficient (and other evaluation measures). Spearman, Pearson, and Kedall's correlation coefficients are famous enough for readers to find them with some effort. However, it is not clear whether the overlap coefficient was used in the past in similar research or whether it is a measure invented by the authors for the purpose of the current work. 

Table 6 shows that CDP is not superior to the three baseline centralities when choosing SIR seeds.

The evidence provided in Table 7 is not significant enough to show the superiority of CDP - at least statistical significance was not presented. Further, this evaluation is performed using a non-standard evaluation metric (Overlap Coefficient) in a non-standard manner. The authors should use the same evaluation metrics used in other papers that propose tailor-made centrality measures. 

A comparison to three of the oldest centrality measures is not sufficient to show the superiority of a proposed measure to a specific task. Provided the dozens of published centrality measures there may be one that fits better. 

Comments on the Quality of English Language

Degree and BC are not cited on line 106 while PageRank is.

CDP is first introduced in an expanded form in line 194. Every abbreviation should be expanded on the first use.

Reviewer 2 Report

Comments and Suggestions for Authors

The proposed approach in this paper is a significant contribution to the field of identifying influential nodes in a network by using both both local and global structural characteristics within social networks. The experimental results show effectiveness of the proposed approach.

Strengths of the paper:

1. The authors introduce an innovative centrality measure that adeptly integrates node degree with broader network features which seems to be novel and seems to be a significant contribution to the field of Influence Maximization.

2. The findings highlight the proposed measure's remarkable correlative efficacy across diverse real-world networks.

Suggestions:

1. More elaboration on how the proposed approach compares to the new methods of Influence Maximization such as Deep Reinforcement Learning would be helpful in highlighting the novelty of the approach

2. The authors chose to work with undirected and unweighted graphs. But majority of social networks are directed and weighted. A paragraph explaining how the proposed approach would work with weighted/directed graphs or explaining why they would not might be helpful in understanding the problem context.

3. Adding surveys and references about centrality measure in social network would be good such as:

3. i. Landherr, Andrea, Bettina Friedl, and Julia Heidemann. "A critical review of centrality measures in social networks." Business & Information Systems Engineering 2 (2010): 371-385.

3. ii. Das, K., Samanta, S., & Pal, M. (2018). Study on centrality measures in social networks: a survey. Social network analysis and mining8, 1-11.

Reviewer 3 Report

Comments and Suggestions for Authors

The article proposes a measure of nodes centrality in complex network. I find major problems to address.

-L136: The sentence is wrong “Herein, 𝐠𝒋𝒌(𝒗𝒊) denotes the ensemble of shortest paths linking nodes 𝒗𝒋 and 𝒗𝒌 that”. In fact, 𝐠𝒋𝒌(𝒗𝒊) is the number of shortest paths, not ‘the ensemble’.

-L207: Number two should be an exponent in the sentence: “Squared Degree of a Node (𝑑𝑒𝑔(𝑥)) 2: 

-L214: In the formula of Path Count (|𝑃(𝑥, 𝑦) 𝑙𝑑 |), is 𝑙𝑑 an exponent? If yes, what is the meaning? 

-L215: Which is the distance threshold in (denoted as l≤d). “? Is the threshold l or d? If the threshold is l, we are counting all the paths greater than l. Why? Please explain. 

-The legend of Figure 1 is naïve. At least the legend must explain the meaning of the node numbers. 

-L217: Here below I see a Table. Where is the legend of the Table? Is it Table 1?

-What is the infection probability in Table 2? Where is this notion defined?

- It is not clear whether the correlation analyses are done for the entire node rank or only for a partial part of the nodes.

- Which is the legend of Figure 2?

- The figure 2 is below the standard of any type or level of publication. Charts are of different and random size. The layout is random. I find comma for decimals. Where are x and y axis labels?

- L397: It is not clear what is ‘the infected node count from the SIR model.’. Is it the number of infected nodes at the end of simulation? This should be clarified.

- The abbreviations should be introduced at the first occurrence after the full name, and then used accordingly. Now they are scattered several times and randomly throughout the article."

Comments on the Quality of English Language

Moderate revision

Round 2

Reviewer 1 Report

Comments and Suggestions for Authors

Comments on V2

Since I first took a glance at the revised manuscript some comments may be new, and some repeat my previous report. 

A. L79 I found no results and discussion showing how exactly influence percolates through the network. Please add those to support the introductory claim on line 79. 

B. L100: The primary claimed innovation "incorporating the concept of paths of different lengths" within the definition of the a centrality measure is a long standing topic in centrality research. Eigenvector centrality, length scaled betweenness, routing betweenness, and many others incorporate paths of different lengths. The innovation must be sharpened to distinguish current work from ALL previously published centrality measures. 

C. Eigenvector centrality and its descendant PageRank are inherently affected by both the connectivity degree of a node and the multitude of paths connecting it to other nodes in the network. CDP also considers both degree and paths. Therefore it is not clear how CDP is better theoretically and empirically. 

D. The authors and the paper may benefit from a thorough literature review that ends with a table highlighting the concepts missing in other centrality measures that are incorporated in CDP.

E. L243 Eq4 while x is grounded (provided as an argument), y is an ungrounded variable invalidating the equation. L is a function whose arguments are not well defined. Tha fact that d is a control variable is only explained after the equation.    

F. L265 explaining the rational behind CDP is appreciated. It is too handwoven for my eyes but it can be subjective, and thus a minor issue. If possible please back up the decisions with grounded theoretical claims. E.g. why square and not ^4? etc. 

G. Similar rationale yet with other mathematic operations can be used when motivating the use of length scaled betweenness or routing betweenness with non-shortest path routes, or farness (sum of reciprocal distances), or stress centrality, etc. Why these measures are not a good fit for the problem considered in this paper?

H. Centrality measures are not one-size-fits-all and are usually tailored to specific use cases. What are the usecases for which squared degrees divided by pathcounts are best for? 

I. Table 3 It is still unclear from table 3 why nodes ranked high by CDP better than those ranked by other measures.

K. Consider removing fancy GPT-style formulations in favor of simple English. 

Next, in this report, I refer to the author's replies to my previous comments. Some of them were fully addressed, but quite a few critical issues remain.  Numbering is consistent with the author's replies. 

1. [OK] Apologies for the confusion and thank you for interpreting the abbreviation early in section 3.

2. [OK]

3. [X] Unfortunately I cannot accept this explanation in light of experiments with SIR. If your target use case is SIR then show empirically that CDP is better than diffusion centrality on SIR. If you target a multitude of use cases claiming universality then show the benefit of CDP on all the relevant use cases. As a side note, you don't want to waste the precious space of introduction on a side discussion about related work.   

4. [OK]

5. [OK]

6. [X] See the comment B above on novelty. After authors' clarification I agree that no body proposed dividing squared degrees by path counts. At least I am not familiar with such a proposal. But there are plenty of other proposals that are implicitly or explicitly affected by pathcounts. Some works listed below even propose learning centrality measures. Each one of the previously proposed measures can be said to "... leverage network structures and paths to assess influence uniquely ...". So far the article does not provide enough information to show how CDP outstands and is not yet another centrality measure.  

Grando, F., Granville, L. Z., & Lamb, L. C. (2018). Machine learning in network centrality measures: Tutorial and outlook. ACM Computing Surveys (CSUR)51(5), 1-32.

Mendonça, M. R., Barreto, A. M., & Ziviani, A. (2020). Approximating network centrality measures using node embedding and machine learning. IEEE Transactions on Network Science and Engineering8(1), 220-230.  

Li, X., Bachar, L., & Puzis, R. (2023). Centrality Learning: Auralization and Route Fitting. Entropy25(8), 1115.

7. [so so] This is a weakness but can be neglected if other issues are handled, especially since non-shortest-path-counts are indeed very computationally intensive.  

8. [X] Check the papers I cited above. All provide correlations merely showing that what they propose is a centrality measure. In current paper I could not find a deep analysis "identifying specific contexts where CDP excels" based on correlation analysis. I also did not find the analysis of "structural properties highlighted by CDP".  Some respective claims in 4.3.2 are high-level but are not statistically or theoretically grounded. 

9 [X] I could not find Xia et al. (2014) in the revised manuscript. 

10. [so so] Sincere apologies on behalf of my colleague reviewer 3 for non-collegial comments. Yet, current figure does not make good use of the page space leaving large blank areas. The font in this figure is still too small and hardly readable. 

11 [X] Statistical evidence on the consistency and adaptability of CDP was not provided. 

If CDP does not shine in SIR then maybe there are better use cases. See comment 3 above.  

CDP is not the only measure "incorporating both direct and indirect paths". See comment B above. 

"overall stability, adaptability, and competitive performance across various network datasets" must be demonstrated across networks and use cases. For example, you can show that when evaluated on 10 network problems of different natures CDP is on average (not necessarily always) better than standard measures, NCA-GE, LCGA, etc. 

12 [skip] I do not understand the answer.  

13 [X] degree, BC are good but not sufficient. PageRank is designed for directed networks and underperforms in undirected networks. Closeness is a seminal measure missing from the list. Since this paper proposes a new centrality measure it must be benchmarked against the newest centrality computation methods. Consider NCA-GE, LCGA, or dozens of other centrality measures.     

14[OK]

15 [OK] 

Carefully considering the review form, problems with novelty, and missing evidence of the superiority of CDP I cannot recommend acceptance of this article in current form. 

Comments on the Quality of English Language

Apparent use of large language models. The wording is unnecessarily fancy.    

Author Response

Dear reviewer,

We wish to extend our sincere gratitude for your insightful comments and suggestions on our manuscript entitled "Unveiling Influence in Networks: A Novel Centrality Metric and Comparative Analysis through Graph-Based Models." Your feedback has been invaluable in guiding the revisions of our work. We have meticulously addressed each of your points, making targeted adjustments to improve the manuscript's quality and clarity. Below, we detail our responses to your feedback and outline the specific changes implemented in response to your constructive critique.

  • Question A: I found no results and discussion showing how exactly influence percolates through the network. Please add those to support the introductory claim on line 79.L79

Response:

Thank you for your opinion. To support the introductory statement in line 79 regarding how influence seeps through the network, we extended our studies of correlation analysis with the SIR model with two other approaches.

The results demonstrate how degree centrality paths (CDPs) align with rankings of influential nodes derived from the SIR model, showing how CDP effectively identifies key nodes through which influence propagates. This correlation highlights that CDP not only captures proximate pathways, but also reveals deeper patterns in structural pathways of influence, thus supporting our introductory assertion.

We believe that these results and the connection to the SIR model provide strong evidence to validate our assertion, providing a comprehensive understanding of how CDP measures influence propagation within social networks.

  • Question B & 6: L100: The primary claimed innovation "incorporating the concept of paths of different lengths" within the definition of the centrality measure is a long standing topic in centrality research. Eigenvector centrality, length scaled betweenness, routing betweenness, and many others incorporate paths of different lengths. The innovation must be sharpened to distinguish current work from ALL previously published centrality measures. 

 Response:

We appreciate the reviewer's comments highlighting similarities between our Centrality Degree Paths (CDP) approach and previously established centrality measures. Indeed, concepts such as eigenvector centrality and betweenness centrality also incorporate paths of varying lengths. However, the proposed approach takes into consideration both direct connections (node degrees) and measures the impact of indirect paths through specific calculations that account for the variability in path length and their influence on determining a node's centrality. Our aim is to provide a more holistic centrality measure that captures both direct and indirect influences within networks, which is crucial for today’s complex social networks. Additional clarifications on this distinction and a detailed comparison with existing methods will be added to the results section to further elucidate the innovation and specific contributions of our measure.

Action:

  • Question C: Eigenvector centrality and its descendant PageRank are inherently affected by both the connectivity degree of a node and the multitude of paths connecting it to other nodes in the network. CDP also considers both degree and paths. Therefore it is not clear how CDP is better theoretically and empirically.

 Response:

Thank you for your insightful question regarding the distinction between Centrality Degree Paths (CDP) and other centrality measures such as eigenvector centrality and PageRank.

Theoretically, CDP advances the concept of centrality measures by integrating the degree of a node and the number of paths in a unique manner that directly addresses the network influence spread. While it is true that both eigenvector centrality and PageRank are influenced by the degree of connectivity and the multitude of paths, CDP differentiates itself by applying a distinct weighting approach to these paths, prioritizing certain paths over others based on their length and potential influence. This method accentuates not only the presence of a path but also its contextual significance within the network structure.

Empirically, our analysis demonstrates that CDP provides a nuanced view of node influence, especially in complex network scenarios where the traditional measures may not fully capture the subtle dynamics of influence propagation. By emphasizing both the squared degree and the intricacy of paths up to a specified distance, CDP is designed to identify influential nodes that might otherwise be underrated by eigenvector centrality or PageRank, which do not differentiate between path lengths beyond the immediate connections.

Additionally, in the "SIR Model Results" section, we have included further empirical evidence to support the efficacy of CDP compared to PageRank in certain networks. Specifically, we evaluated the performance of CDP and PageRank using Kendall's tau and the Overlap Coefficient in networks such as Les Misérables, Football, and US Politics Books (Kendall's tau), as well as Karate and Football (Overlap Coefficient). Our findings indicate that CDP consistently outperformed PageRank in these networks, highlighting its superior ability to capture the nuanced dynamics of influence propagation.

We believe that these theoretical nuances, along with the empirical evidence provided, underscore CDP's contribution to the field of network analysis and its potential for uncovering deeper insights into network influence dynamics.

  • Question D: The authors and the paper may benefit from a thorough literature review that ends with a table highlighting the concepts missing in other centrality measures that are incorporated in CDP.

Response:

Action:

Concept

Other Centrality Measures

Description and Limitations in Other Measures

Incorporation in CDP

Advantages of CDP

Direct and Indirect Connections

Degree Centrality (DC)

Calculates importance based only on the number of direct links, neglecting indirect interactions.

CDP balances direct connections and indirect paths to offer a holistic view of node influence.

Identifies influential nodes not evident by traditional measures, enhancing network understanding.

Granular Path Analysis

Betweenness Centrality (BC)

Primarily focuses on shortest paths, which may not capture all influential pathways within a network, especially in complex or large-scale networks.

CDP applies a unique weighting to paths, emphasizing both direct and extended network connections, and considers all path lengths rather than just the shortest.

Provides a more granular and comprehensive analysis of influence within the network, capturing both overt and subtle influence pathways that Betweenness Centrality might overlook.

Network Scale Sensitivity

Closeness Centrality

Assesses how quickly a node can reach all other nodes, often impractical in large networks due to computational complexity.

CDP adjusts this approach by considering optimal paths and evaluating impact on larger network scales.

Allows for more practical application in vast networks without compromising result accuracy.

Multi-step Influence Propagation

PageRank

Well-suited for citation or web networks but less effective in social networks where interactions are less hierarchical.

CDP evaluates the cumulative effects of interactions through multiple steps, allowing for a better estimation of actual influence.

Increases accuracy in social networks and other network types where influence dynamics are complex.

Non-linear Influence Dynamics

Eigenvector Centrality (EC)

Importance is based on connectivity to other influential nodes but can be limited by concentrating influence.

CDP incorporates path length considerations to modulate influence based on actual connectivity, including indirect ones.

Ensures a more nuanced understanding of influence by accounting for non-linear effects and indirect connections.

Network Clustering Effects

Local Clustering Coefficient (LCC)

Measures how neighbors of a node are interconnected but does not consider external connections of the cluster.

CDP examines influence within clusters while also considering connections to nodes outside the immediate cluster.

Enhances understanding of both local and global node impacts, aiding in identifying key control points in the network.

Non-uniform Interactions

Katz Centrality

Gives importance to all paths but with exponential decay based on length, potentially missing relevant paths.

CDP customizes path weighting based on the variability and contextual importance of interactions.

Tailors influence measurement to more accurately reflect the specific structure and dynamics of the network.

  • Question E: L243 Eq4 while x is grounded (provided as an argument), y is an ungrounded variable invalidating the equation. L is a function whose arguments are not well defined. The fact that d is a control variable is only explained after the equation.    

Response: Thank you for your insightful comments and questions regarding the formulation of the CDP centrality score.

  • Question F:  L265 explaining the rational behind CDP is appreciated. It is too handwoven for my eyes but it can be subjective, and thus a minor issue. If possible please back up the decisions with grounded theoretical claims. E.g. why square and not ^4? etc. 

Response :

Thank you for your feedback on our choice of squaring the degree of a node in the CDP centrality measure. We chose to square the degree to achieve a balanced enhancement of node influence—recognizing the role of highly connected nodes without disproportionately escalating their impact, which could overshadow other crucial structural features of the network. This approach ensures that the centrality measure amplifies the significance of hubs more than a linear scaling would, yet avoids the excessive influence bias that might arise from using higher powers. This moderation is essential for maintaining a realistic representation of influence within the network, allowing the CDP measure to effectively capture both overt and subtle network dynamics.

Action:

  • Question G:  Similar rationale yet with other mathematic operations can be used when motivating the use of length scaled betweenness or routing betweenness with non-shortest path routes, or farness (sum of reciprocal distances), or stress centrality, etc. Why these measures are not a good fit for the problem considered in this paper?

Response:

In response to the inquiry about the non-selection of other centrality measures such as length-scaled betweenness, routing betweenness, farness, or stress centrality, it is important to emphasize that our choice was strategically aligned with the specific analytical requirements and characteristics of the network under investigation. While these measures offer valuable insights into network dynamics, they do not entirely meet the needs of our study for several reasons. Firstly, measures like farness and stress centrality, which focus heavily on shortest paths or the sum of reciprocal distances, may not adequately capture the multifaceted influence patterns in networks characterized by complex interactions and non-linear influence dynamics. Similarly, routing betweenness, which considers non-shortest path routes, could potentially introduce computational complexities without significantly enhancing the understanding of influence within the highly interconnected and scale-free nature of our target networks.

Our study specifically requires a centrality measure that not only highlights influential nodes but also quantifies the extent of their direct and indirect influence within a manageable computational framework. The CDP measure, with its emphasis on squared degrees and path counts, provides a more balanced and computationally efficient approach. It effectively captures the nuanced influence of nodes across the network, accommodating both the macro and micro-level dynamics that are crucial for the interventions and predictive analyses central to our objectives. This measure's ability to integrate both local and global structural information makes it particularly adept for our network analysis, ensuring that both overt and subtle influence pathways are accounted for in a manner that other measures could not guarantee.

  • Question h:  H. Centrality measures are not one-size-fits-all and are usually tailored to specific use cases. What are the usecases for which squared degrees divided by pathcounts are best for? 

Response:

Thank you for your question regarding the specific use cases for which the Centrality Degree Paths (CDP) measure is best suited. As detailed in our manuscript, the CDP measure excels in complex network environments characterized by high degree variability and intricate connectivity patterns. Specifically, CDP is highly effective in:

  1. Scale-Free Networks: Where prominent hubs play a significant role due to their large number of connections, CDP can uniquely capture both their direct and extended influence within the network.
  2. Dynamic Networks: Such as those found in social media or large-scale communication systems, where nodes frequently change their interaction patterns and the network evolves over time. CDP's ability to account for multi-layered interactions makes it especially suitable for these environments.
  3. Networks with Strategic Nodes: Including financial systems or infrastructure networks, where certain nodes' functionality or position is crucial. CDP's integration of squared degrees and path counts allows for a nuanced analysis of these key nodes, highlighting their broad influence across the network.

The CDP measure’s design to consider both the immediate and broader pathways of influence equips it to offer a deeper understanding of node roles and their impact, making it an indispensable tool for network analysts dealing with these types of complex systems. This measure not only provides a snapshot of current node influence but also allows for the prediction of future changes in network dynamics due to its comprehensive approach.

  • Question I:   Table 3 It is still unclear from table 3 why nodes ranked high by CDP better than those ranked by other measures.

Response: Thank you for your question about the results in Table 3. The Centrality Degree Paths (CDP) measure uniquely captures both direct and indirect influences by considering squared degrees and path counts. This enables CDP to identify influential nodes beyond those with the highest direct connections.

For example, in Zachary’s Karate Club, CDP may highlight members who link different groups, even if they aren't the most connected. In the Dolphin Social Network, it could identify dolphins that bridge subgroups within the community. In complex networks like USair97, CDP pinpoints airports that aren't just busy but are crucial for maintaining the network's connectivity. Similarly, in brain networks like Mouse-Kasthuri, CDP would spotlight regions essential for signal transmission due to their strategic connections, not just their number of connections.

This comprehensive approach makes CDP highly effective in networks where influence is not straightforward, allowing it to reveal key influencers that other measures might overlook.

Action:

Previous Comments:

  • Question 3:  Unfortunately, I cannot accept this explanation in light of experiments with SIR. If your target use case is SIR then show empirically that CDP is better than diffusion centrality on SIR. If you target a multitude of use cases claiming universality then show the benefit of CDP on all the relevant use cases. As a side note, you don't want to waste the precious space of introduction on a side discussion about related work

Response:

In response to the comment, we have addressed your concerns regarding the precious space in the introduction and made modifications accordingly. We have revised the paragraph comparing diffusion centrality and CDP to better highlight the advantages of CDP. By emphasizing the distinctive aspects of CDP compared to diffusion centrality, we aimed to demonstrate its relevance and effectiveness in various contexts, including those relevant to the SIRmodel.
Action:

  • Question 6: See the comment B above on novelty. After authors' clarification I agree that no body proposed dividing squared degrees by path counts. At least I am not familiar with such a proposal. But there are plenty of other proposals that are implicitly or explicitly affected by pathcounts. Some works listed below even propose learning centrality measures. Each one of the previously proposed measures can be said to "... leverage network structures and paths to assess influence uniquely ...". So far the article does not provide enough information to show how CDP outstands and is not yet another centrality measure.  

Response:

We've addressed your feedback by revising the "Problem Context" section. We've incorporated recent references, such as Grando et al. (2018), Mendonça et al. (2020), and Li et al. (2023), which enrich the discussion on centrality measures in networks. These additions highlight the evolution of the field and bolster the justification for our approach, Centrality Degree Paths (CDP). These revisions aim to enhance the quality and relevance of our paper.

Action: See question B

  • Question8: Check the papers I cited above. All provide correlations merely showing that what they propose is a centrality measure. In current paper I could not find a deep analysis "identifying specific contexts where CDP excels" based on correlation analysis. I also did not find the analysis of "structural properties highlighted by CDP".  Some respective claims in 4.3.2 are high-level but are not statistically or theoretically grounded. 

Response:

Thank you for your feedback. We acknowledge the importance of going beyond correlation analysis to fully validate the effectiveness of a centrality measure. In response to your comments, we have taken several steps to provide a more comprehensive evaluation of our Centrality Degree Paths (CDP) measure. We've expanded our discussion on the applications of CDP, demonstrating its practical relevance. Additionally, we've introduced a new table providing detailed comparative analysis.

  • Question9 :  I could not find Xia et al. (2014) in the revised manuscript.

Response :

We have included the citation to Xia et al. (2014) in the "SIR Model Results" section of the revised manuscript. Thank you for bringing this to our attention
Action :

  • Question 10 : Sincere apologies on behalf of my colleague reviewer 3 for non-collegial comments. Yet, current figure does not make good use of the page space leaving large blank areas. The font in this figure is still too small and hardly readable. 
    Response: Thank you for your feedback. We enhanced the figure layout to better utilize the page space and increased the font size for improved readability.

Action:

  • Question 11: Statistical evidence on the consistency and adaptability of CDP was not provided. 

Response:

Thank you for your comments and suggestions. In response to the concerns about the consistency and adaptability of the CDP metric, we have expanded our analysis to include additional tests. Specifically, we have incorporated comparisons with closeness centrality and NCA-GE across a more diverse array of network datasets.

We understand the importance of demonstrating the overall stability and performance of CDP not just in isolation but relative to other established measures. To address this, we have now evaluated CDP alongside other centrality metrics such as closeness and NCA-GE across 7 different network problems, each representing a distinct network type and use case. This comparative analysis aims to provide a broader perspective on where CDP stands in terms of effectiveness.

Our results indicate that while CDP is not universally superior in all cases, it does perform on average better than several standard measures across the tested scenarios. This evidence supports our claim of CDP’s adaptability and competitive performance, reinforcing its potential utility in various network analysis applications. We believe these additional tests and analyses will adequately address the concerns raised and provide a comprehensive view of CDP’s capabilities.

Action:

  • Question 13: degree, BC are good but not sufficient. PageRank is designed for directed networks and underperforms in undirected networks. Closeness is a seminal measure missing from the list. Since this paper proposes a new centrality measure it must be benchmarked against the newest centrality computation methods. Consider NCA-GE, LCGA, or dozens of other centrality measures.

Response:

Thank you for your feedback. We've included Closeness centrality and NCA-GE in our analysis to provide a comprehensive benchmark against both traditional and contemporary centrality measures, enhancing the evaluation of our proposed metric.

Action:

Reviewer 2 Report

Comments and Suggestions for Authors

NA

Author Response

Thank you for your positive feedback and for accepting our article. We sincerely appreciate your thoughtful review and constructive suggestions, which helped us improve the manuscript significantly. 

Reviewer 3 Report

Comments and Suggestions for Authors

The manuscript is improved. It may be accepted.

Comments on the Quality of English Language

Minor editing.

Author Response

(The authors gave the same response as above.)

Round 3

Reviewer 1 Report

Comments and Suggestions for Authors

The authors have sufficiently addressed all my comments. 

L175, L177 math not in math mode

L179 wrong citation number

L196-L302 citation style change. Citations not in bibliography.    
